# ACTOR: Active Learning with Annotator-specific Classification Heads to Embrace Human Label Variation

**Xinpeng Wang** and **Barbara Plank**

MaiNLP, Center for Information and Language Processing, LMU Munich, Germany
Munich Center for Machine Learning (MCML), Munich, Germany
{xinpeng, bplank}@cis.lmu.de

## Abstract

Label aggregation such as majority voting is commonly used to resolve annotator disagreement in dataset creation. However, this may disregard minority values and opinions. Recent studies indicate that learning from individual annotations outperforms learning from aggregated labels, though they require a considerable amount of annotation. Active learning, as an annotation cost-saving strategy, has not been fully explored in the context of learning from disagreement. We show that in the active learning setting, a multi-head model performs significantly better than a single-head model in terms of uncertainty estimation. By designing and evaluating acquisition functions with annotator-specific heads on two datasets, we show that group-level entropy works generally well on both datasets. Importantly, it achieves performance in terms of both prediction and uncertainty estimation comparable to full-scale training from disagreement, while saving 70% of the annotation budget.

## 1 Introduction

An important aspect of creating a dataset is asking for multiple annotations and aggregating them in order to derive a single *ground truth* label. Aggregating annotations, however, implies a single golden ground truth, which is not applicable to many subjective tasks such as hate speech detection (Ovesdotter Alm, 2011). A human's judgement on subjective tasks can be influenced by their perspective and beliefs or cultural background (Waseem et al., 2021; Sap et al., 2022). When addressing disagreement in annotation, aggregating them by majority vote could result in the viewpoints of the minority being overlooked (Suresh and Guttag, 2019).

In order to address this issue, many works have been proposed to directly learn from the annotation disagreements in subjective tasks. There are two major approaches to achieving that: learning from the *soft label* (Peterson et al., 2019; Uma et al.,

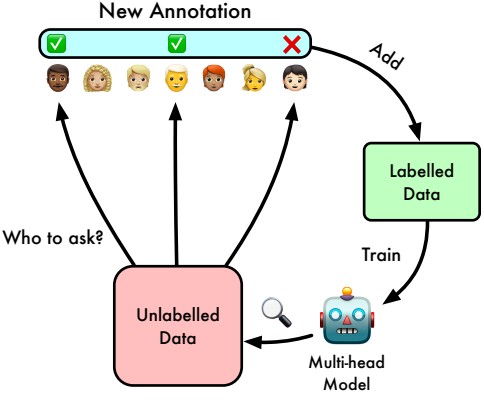

Figure 1: For each sample that needs to be labelled, our model actively selects specific annotators for annotations to learn from the label variation.

2020; Fornaciari et al., 2021) and learning from the *hard label* of individual annotators (Cohn and Specia, 2013; Rodrigues and Pereira, 2018; Davani et al., 2022).

In a recent work, Davani et al. (2022) shows that modelling the individual annotators by adding annotator-specific classification heads in a multi-task setup outperforms the traditional approach that learns from a majority vote. However, training such a model needs a huge amount of data with multiple annotations to model the opinions and beliefs of the individual annotators.

On another line, Active Learning (AL) is a framework that allows learning from limited labelled data by querying the data to be annotated. In this paper, we propose to take the best of both worlds: active learning and human label variation, to mitigate the high cost of the annotation budget needed for training the model. In particular, we propose a novel active learning setting, where the multi-head model actively selects the annotator and the sample to be labelled. Our results show this effectively reduces annotation costs while at the same time allowing for modelling individual perspectives.

**Key Findings** We made several key observations:

- The multi-head model works significantly better than the single-head model on uncertainty estimation.

- The use of group-level entropy is generally recommended. Individual-level entropy methods perform differently depending on the dataset properties.

- The multi-head model achieves a performance comparable to full-scale training with only around 30% annotation budget.

## 2 Related Work

### 2.1 Learning from Disagreement

There is a growing body of work that studies irreconcilable differences between annotations (Plank et al., 2014; Aroyo and Welty, 2015; Pavlick and Kwiatkowski, 2019; Uma et al., 2021). One line of research aims at resolving the variation by aggregation or filtering (Reidsma and Carletta, 2008; Beigman Klebanov et al., 2008; Hovy et al., 2013; Gordon et al., 2021). Another line of research tries to embrace the variance by directly learning from the raw annotations (Rodrigues and Pereira, 2018; Peterson et al., 2019; Fornaciari et al., 2021; Davani et al., 2022), which is the focus of our paper.

### 2.2 Active Learning

In active learning, many different methods for selecting data have been proposed to save annotation cost, such as uncertainty sampling (Lewis, 1995) based on entropy (Dagan and Engelson, 1995) or approximate Bayesian inference (Gal and Ghahramani, 2016). Other approaches focus on the diversity and informativeness of the sampled data (Sener and Savarese, 2017; Gissin and Shalev-Shwartz, 2019; Zhang and Plank, 2021). Herde et al. (2021) proposed a probabilistic active learning framework in a multi-annotator setting, where the disagreement is attributed to errors. Recent work by Baumler et al. (2023) accepted the disagreement in the active learning setting, and they showed improvement over the passive learning setting using single-head model. Our work shows the advantage of the multi-head model and compares it with traditional single-head active learning methods.

## 3 Method

**Multi-head Model** We use a multi-head model where each head corresponds to one unique annotator, following Davani et al., 2022. In the fine-tuning stage, annotations are fed to the corresponding annotator heads, adding their losses to the overall loss. During testing, the F1 score is calculated by comparing the majority votes of the annotator-specific heads with the majority votes of the annotations.

### 3.1 Multi-head Acquisition Functions

We study five acquisition functions for the multi-head model. Since our model learns directly from the annotation, we care about which annotator should give the label. So we query the instance-annotation pair $(x_i, y_i^a)$ with its annotator ID $a$. In this way, our data is duplicated by the number of annotations available.

**Random Sampling (Rand.)** We conduct random sampling as a baseline acquisition method where we randomly sample $K$ (*data, annotation, annotator ID*) pairs from the unlabeled data pool $U$ at each active learning iteration.

**Individual-level Entropy (Indi.)** Intuitively, the annotator-specific heads model the corresponding annotators. Therefore, we can calculate the entropy of the classification head to measure the specific annotator's uncertainty. Given the logits $z^a = [z_1^a, ..., z_n^a]$ of the head $a$, the entropy is calculated as following: $H_{indi}(p^a|x) = -\sum_{i=1}^n p_i^a(x) \log(p_i^a(x))$, where $p_i^a(x) = \text{softmax}(z_i^a(x))$. Then we choose the (*instance, annotator*) pair with the highest entropy: $\text{argmax}_{x \in U, a \in A} H_{indi}(p^a|x)$, where $U$ denotes the unlabeled set and $A$ denotes the annotator pool. We compute entropy only for the remaining annotators who have not provided annotations for the instance.

**Group-level Entropy (Group)** Instead of looking at the individual's uncertainty, we can also query the data by considering the group-level uncertainty. One way to represent the uncertainty of the group on a sample is to calculate the entropy based on the aggregate of each annotator-specific head's output. Therefore, we normalize and sum the logits of each head at the group level: $z_{group} = [z_1, ..., z_n] = \sum_{h=1}^H z_{norm}^h$, and calculate the group-level entropy as follows: $H_{group}(x) = -\sum_{i=1}^n p_i(x) \log(p_i(x))$, where $p_i(x) = \text{softmax}(z_i(x))$. We then query the data with the highest uncertainty.

**Vote Variance (Vote)** Another way to measure the uncertainty among a group is by measuring

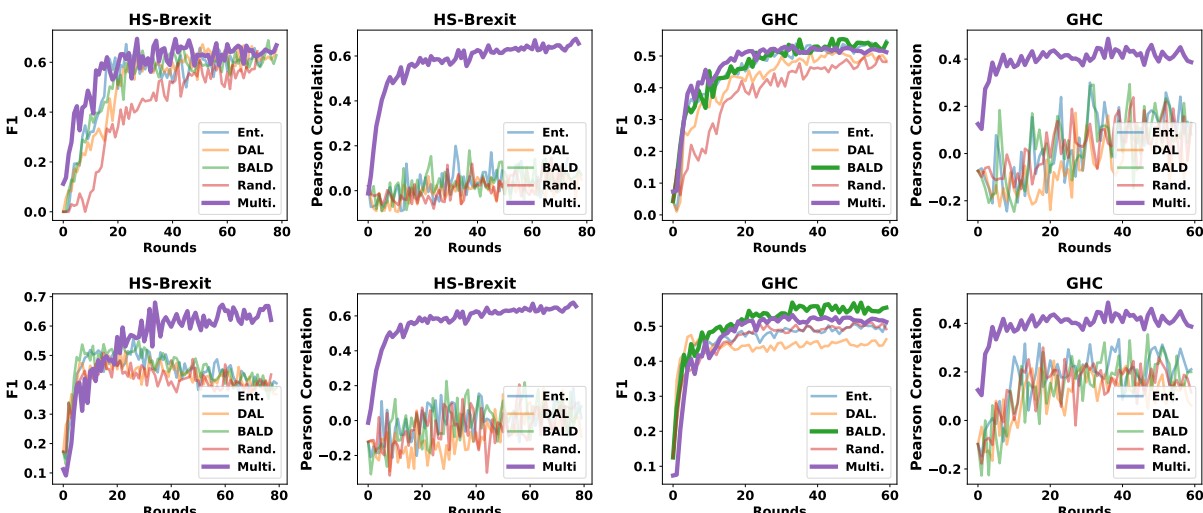

Figure 2: Comparsion of the Multi-head model and Single-Majority (upper row) and Single-Annotation (bottom row). Results are averaged over 4 runs. All the methods have the same annotation cost of the seed dataset and the queried batch at each round.

the variance of the votes. Given the prediction $y^h$ of classification head $h$, we calculate the vote variance: $\text{Var} = \frac{1}{H} \sum_{i=1}^{H} (y^h - \mu)^2$, where $\mu = \frac{1}{H} \sum_{h=1}^{H} y^h$. This approach can be applied to binary classification or regression problems.

**Mixture of Group and Indi. Entropy (Mix.)** We also consider a variant which combines the group-level and individual-level entropy by simply adding the two: $H_{mix} = H_{indi} + H_{group}$.

## 4 Experiments

**Dataset** We selected two distinct hate speech datasets for our experiments: **Hate Speech on Brexit** (**HS-Brexit**) (Akhtar et al., 2021) and **Gab Hate Corpus** (**GHC**) (Kennedy et al., 2022). We split the raw annotation dataset according to the split of the aggregated version dataset provided. The **HS-Brexit** dataset includes 1,120 English tweets relating to Brexit and immigration, where a total of six individuals were involved in annotating each tweet. As each tweet contains all six annotations, we refer to HS-Brexit as *densely* annotated. In **GHC**, 27,665 social-media posts were collected from the public corpus of Gab.com (Gaffney, 2018). From a set of 18 annotators, each instance gets at least three annotations. Therefore, GHC is *sparsely* annotated. Both datasets contain binary labels $y \in [0, 1]$ and have almost the same positive/negative raw annotation ratio (0.15).

**Single-head Model Baselines** We implement four acquisition methods for single-head model active learning for comparison: Random sampling

(**Rand.**), Max-Entropy (**Ent.**; Dagan and Engelson, 1995), Bayesian Active Learning by Disagreement (**BALD**; Houlsby et al., 2011) and Discriminative Active Learning (**DAL**; Gissin and Shalev-Shwartz, 2019). We compare them with the multi-head approach with random sampling which has an average performance among the five multi-head acquisition methods we investigated.

Two different single-head model approaches are considered: Learning from the Majority Vote (**Single-Majority**) and Learning from Raw annotations (**Single-Annotation**). In the first setting, all annotators' annotations are queried, and the majority vote is used to train the model. In the second setting, we train the model with individual annotations without aggregation, following the repeated labelling approach by Sheng et al. (2008).

**Experimental Setup** We follow the setup of Davani et al. (2022) for modelling and evaluation. We initialize the layers before the heads with a BERT-base model (Devlin et al., 2019). To balance training data, we do oversampling following Kennedy et al. (2022). Moreover, we use class weights on the loss function for multi-head model training, which makes it more stable. It is not used for the single-head model as it degrades performance.

To evaluate the model, we first report the F1 score against the majority vote. Secondly, we also compute individual F1 scores, measuring annotator-specific heads against annotator labels. Thirdly and importantly, we are interested to gauge how well the model can predict the data uncertainty

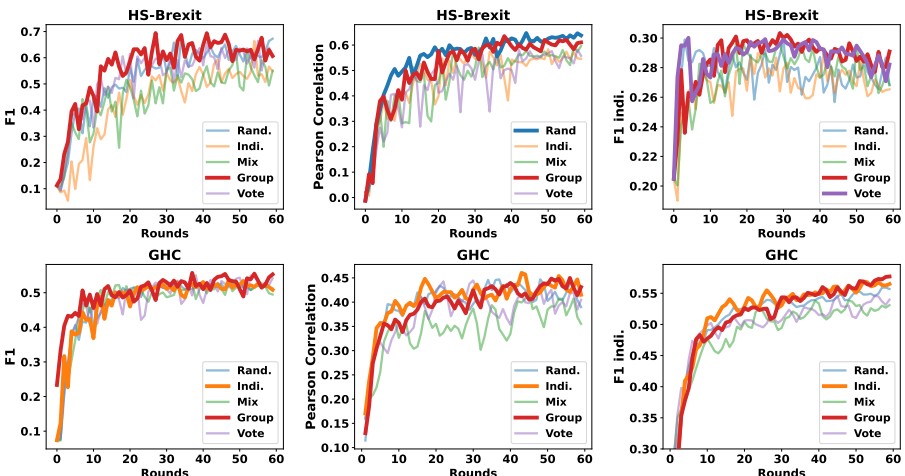

Figure 3: Comparison of multi-head acquisition functions. Results are averaged over 4 runs. Group-level entropy method (**Group**) performs generally well on both datasets on all three metrics. Individual-level uncertainty (**Indi.**) only performs well on GHC.

by calculating the Pearson correlation between the model's uncertainty and the annotation disagreement measured by the variance of the annotations on the same instance. For the single-head model, we use *Prediction Softmax Probability* proposed by Hendrycks and Gimpel (2017) as the uncertainty estimation of the model. For the multi-head model, we follow Davani et al. (2022) and calculate the variance of the prediction of the heads as the model's uncertainty.

## 5 Result

**Single-head vs Multi-head Model** Figure 2 shows the comparison of the multi-head model and the single-head during the active learning process In the upper row, we compare the *multi-head* approach with *single-majority* approach on majority F1 score and uncertainty estimation. In terms of predicting the majority vote, the *multi-head* model performs on par with the best-performing *single-head* method on both datasets, such as BALD. For uncertainty estimation measured against annotator disagreement, the *multi-head* model outperforms the *single-head* model by a large margin.

We have the same observation when comparing with *single-annotation* model, shown in the bottom row. Therefore, we recommend using a *multi-head* model in a subjective task where humans may disagree and uncertainty estimation is important.

**Label Diversity vs. Sample Diversity** When it comes to *group-level* uncertainty based acquisition functions (**Group** and **Vote**), we tested two approaches to determine which annotator to query

from: *Label Diversity First* and *Sample Diversity First*. In *Label Diversity First*, we query from all the available annotators to prioritize the label diversity of a single sample. In *Sample Diversity Fisrat* approach, we only randomly choose one of the annotators for annotation. Given the same annotation budget for each annotation round, *Label Diversity First* would query fewer samples but more annotations than *Sample Diversity First* approach. In our preliminary result, *Label Diversity First* shows stronger performance in general. Therefore, we adopt this approach for the following experiments.

**Comparison of Multi-head acquisition functions** To compare different strategies to query for annotations, we compare the five proposed acquisition functions from Section 3.1 in Fig 3. **Group** performs generally well on both datasets. We also see a trend here that HS-Brexit favours acquisition function based on *group-level* uncertainty (**Vote**), while *individual-level* uncertainty (**Indi.**) works better on GHC dataset. For HS-Brexit, **Group** is the best-performing method based on the majority F1 score. When evaluated on raw annotation (F1 indi. score), both vote variant and group-level entropy perform well. For uncertainty estimation, random sampling is slightly better than group-level entropy approach. On the GHC dataset, both **Indi.** and **Group** perform well on uncertainty estimation and raw annotation prediction. However, we don't see an obvious difference between all the acquisition functions on the majority vote F1 score.

**Annotation Cost** In terms of saving annotation cost, we see that the F1 score slowly goes into a

plateau after around 25 rounds on both datasets in Fig 3, which is around 30% usage of the overall dataset (both datasets are fully labelled at around 90 rounds). For example, *Vote* achieves the majority F1 score of 52.3, which is 94% of the performance (55.8) of the full-scale training (round 90).

## 6 Conclusion

We presented an active learning framework that embraces human label variation by modelling the annotator with annotator-specific classification heads, which are used to estimate the uncertainty at the individual annotator level and the group level. We first showed that a multi-head model is a better choice over a single-head model in the active learning setting, especially for uncertainty estimation. We then designed and tested five acquisition functions for the annotator-heads model on two datasets. We found that group-level entropy works generally well on both datasets and is recommended. Depending on the dataset properties, the individual-level entropy method performs differently.

## Limitations

The multi-head approach is only viable when the annotator IDs are available during the active learning process since we need to ask the specific annotator for labelling. Furthermore, the annotators should remain available for a period of time in order to provide enough annotations to be modelled by the specific head successfully. Note that we here simulate the AL setup. The multi-head approach is good at estimating the uncertainty based on the annotators it trains on, however, whether the uncertainty can still align with yet another pool of people is still an open question.

Further analysis is needed to understand why GHC and HS-Brexit favour different acquisition functions. Besides the difference between *dense* and *sparse* annotation, factors such as *diversity* of the topics covered and annotator-specific annotation statics are also important, which we leave as future work.

## Acknowledgements

We thank the anonymous reviewers for their feedback. This research is supported by the ERC Consolidator Grant DIALECT 101043235.

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

## A  Data Preparation

In contrast to the traditional active learning approach, which only selects instances from the unlabeled data pool, our approach also considers which annotator should give the annotation since the annotation should be fed to the annotator-specific heads. To address this issue, we reform the dataset by splitting one instance with N annotations into N (data, annotation) pairs with its annotator ID. When selecting the batch from the populated data pool, the (data, annotation) pair is given to the classification head corresponding to its annotator ID.

## B  Active Learning Setting

Since our work focus on raw annotation, we set the seed dataset size and the query batch size based on the annotation budget. We set the annotation budget for HS-Brexit as (60, 60) for both seed data size and query data size respectively. For GHC, we set it as (200, 200).

## C  Training Setup

We list our training parameter in Table 1. We halve the learning rate when the F1 score decreases at evaluation.

| Parameter | Value |
| --- | --- |
| Weight Decay | 0.01 |
| Optimizer | AdamW |
| Adam $\epsilon$ | 1e-6 |
| Adam $\beta_1$ | 0.9 |
| Adam $\beta_2$ | 0.99 |
| Gradient Clipping | 0.0 |
| Peak Learning Rate | 2e-5 |
| Batch size | 32 |
| Early stopping | 5 |

Table 1: Parameter used for training.