# OpenReview forum: "ACTOR: Active Learning with Annotator-specific Classification Heads to Embrace Human Label Variation"
_EMNLP/2023/Conference — EMNLP 2023 Main_

### Official Review · Reviewer_9qCH · 2023-07-25

**Soundness:** 3

**Excitement:**

3: Ambivalent: It has merits (e.g., it reports state-of-the-art results, the idea is nice), but there are key weaknesses (e.g., it describes incremental work), and it can significantly benefit from another round of revision. However, I won't object to accepting it if my co-reviewers champion it.

**Paper Topic And Main Contributions:**

The authors propose using a multi-head model, in which each head models one annotator for active learning. They also present multiple AL methods that should go well with their multi-head model. Authors show their methods are competitive against other popular methods for single-head models. Their model is also better at modeling individual opinions. Lastly, they use the Pearson correlation between the model's uncertainty and the disagreement between annotators and show that their model's uncertainty is by far better at modeling human uncertainty than the single-head models.

**Questions For The Authors:**

A) You claim your method can select which annotators to query. How do you do that if you are using the group-level entropy? Or are you able to select annotators only for individual-level entropy?

B) How do you compare different setups? I see you use the F1 score, but I wonder if you mention at which step is F1 score calculated. I guess it is at the end of the AL loop, but could you confirm that, and if you mentioned that in the text, please correct me.

C) How do you calculate individual F1 scores for the Single-Annotation model? Does that model have access to the information about the label's annotator? If not, that could explain a rather large performance gap between your methods on the Multi-Head model and methods on the Single-Annotation model, as seen in Figure 2. Please comment on why you used the individual F1 score to compare Multi-head and Single-Annotation models in Figure 2. It does not seem fair to use scoring that considers the label's annotators when one model is explicitly trained to model annotators, and another model has no information about the annotators but is just trained on multiple labels for the same instance. Or does the Single-Annotation model have access to the information about the label's annotators during training, and if yes, how?

**Reasons To Accept:**

- The idea is interesting. Letting the model choose not just what instances to label but also what annotators should label them is an exciting approach to active learning.
- Creating models that are not only optimized to model the annotators' opinions but are also good at modeling annotators' confidence is an important research direction.

**Reasons To Reject:**

- In the paragraphs Individual-level entropy and Group-level entropy, authors represent entropy with $H$. It would be better to make it clear that entropy depends on a datapoint, e.g., $H(p^a|x)$ , so that formulas like $\text{argmax}_{x \in U, a \in A} H^a$ became more understandable.
- The authors' formulation of Group-level entropy adds up the logits of all the heads to calculate the entropy. However, since each head uses its own projection to calculate logits, it does not make a lot of sense to add them since they can be on a different scale. For example, if one head's logits are of a much higher magnitude than the logits of the other heads, the larger logits could take over, and the Group-level entropy would equal the entropy of that one head. It would be safer to calculate the joint entropy of the posterior probabilities of the model. If you could assume independence of the heads, it is just the sum of the entropies of each of the heads, although I wonder if it is a valid assumption since we expect to see some agreement between annotators. If independence can not be assumed, it is more challenging to calculate joint entropy since one needs to calculate joint distributions. The problem is that the approach has no theoretical grounding and can be easily overwhelmed and overtaken by a single head. A simple fix could be to normalize the logits of different heads before adding them up.
- In the paragraph Vote variance, authors explain how they calculate vote variance from the predictions of the classification heads. But the given formulation makes sense only for binary classification, i.e., $y^h \in $ {$0,1$}, because calculating the average vote $\mu$ with more than two classes wouldn't make sense since you are averaging categorical variables. Since all other methods are more general, i.e., work with more classes than Vote variance, it would be important to mention it.

**Reproducibility:**

4: Could mostly reproduce the results, but there may be some variation because of sample variance or minor variations in their interpretation of the protocol or method.

**Reviewer Confidence:**

4: Quite sure. I tried to check the important points carefully. It's unlikely, though conceivable, that I missed something that should affect my ratings.

**Typos Grammar Style And Presentation Improvements:**

- Title case the title.

- Line 104 – use \citet{}

- Line 259 – variant -> variance

---

> ### Author Rebuttal · Authors · 2023-08-28
>
> **Q1:  It would be better to make it clear that entropy depends on a datapoint $H\left(p^a \mid x\right)$**
>
> A1: Thanks for this notation suggestion. We will add this in the camera-ready version.
>
> **Q2: Adding up the logits from different heads doesn’t make a lot of sense since they can be on different scales.  A simple fix could be to normalize the logits of different heads before adding them up.**
>
> A2: We appreciate this valuable suggestion. We conducted additional experiments on summing the normalised logits as the group entropy.
> We normalize the logits vector of each annotator heads before adding them up.
> Majority vote F1 score and Pearson Correlation results on HS-BREXIT are listed below.
>
> __Majority vote F1__
> | AL iteration | 0 | 10 | 20 | 30 | 40 | 50 | 60 |
> | :--- | :--- | :--- | :--- | :--- | :--- | :--- | :--- |
> | Group_Normalized | 0.10 | **0.55** | **0.61** | 0.54 | **0.65** | 0.63 | **0.67** |
> | Group | 0.11 | 0.21 | 0.41 | 0.48 | 0.54 | 0.55 | 0.53 |
> | Random | **0.12** | 0.34 | 0.47 | **0.62** | 0.61 | **0.65** | **0.67** |
> | Vote | **0.12** | 0.44 | 0.54 | 0.53 | 0.64 | 0.59 | 0.60 |
>
> __Pearson's Correlation__
> | AL iteration | 0 | 10 | 20 | 30 | 40 | 50 | 60 |
> | :--- | :--- | :--- | :--- | :--- | :--- | :--- | :--- |
> | Group_Normalized | **0.07** | 0.37 | 0.53 | 0.52 | 0.60 | 0.59 | 0.61 |
> | Group | 0.02 | 0.34 | 0.54 | 0.53 | 0.46 | 0.55 | 0.50 |
> | Random | -0.01 | **0.48** | **0.59** | **0.59** | **0.62** | **0.61** | **0.64** |
> | Vote | 0.02 | 0.35 | 0.51 | 0.51 | 0.51 | 0.56 | 0.56 |
>
> __Individual F1__
> | AL iteration | 0 | 10 | 20 | 30 | 40 | 50 | 60 |
> | :--- | :--- | :--- | :--- | :--- | :--- | :--- | :--- |
> | Group_Normalized | 0.20 | **0.30** | **0.30** | **0.31** | 0.30 | **0.31** | **0.30** |
> | Group | **0.21** | 0.27 | **0.30** | 0.30 | 0.30 | 0.28 | 0.29 |
> | Random | **0.21** | 0.28 | 0.28 | 0.29 | 0.28 | 0.28 | 0.27 |
> | Vote | 0.20 | 0.28 | 0.29 | 0.30 | 0.30 | 0.29 | 0.28 |
>
> We do observe improved performance of using the normalised logits compared to the raw logits in all the three metrics . In terms of majority F1 score, it performs the best among all the baselines at the early stage and on par with the Rand. baseline at later stage. In terms of uncertainty estimation, Rand. is still the best performing method. When measured against individual F1, normolized version Group entropy performs the best.  We will include the results as well as the GHS dataset results in the final version.
>
> **Q3:  Vote variance only works when there are only two classes.**
>
> A3: This is a very good observation. We will mention this restrict in the paper since we only study hate speech detection task which only requires two classes. Other metrics should be considered such as disagreement score when there are more classes.
>
> **Q4: How to select  annotators when using group-level entropy?**
>
> A4: We select all the available annotators. In our preliminary experiments, we tried two approaches when using group-level entropy:
> 1. select all the annotators 2. randomly select one annotator.
>
> In our preliminary results, the first approach performs better.
> Our methods only query the specific annotators when using the individual-level entropy as an acquisition function.
> Our results show that individual-level entropy benefits more in GHC dataset than in HS-BREXIT.
> We will make this point clear in the final paper.
>
>
> **Q5: At which step is the F1 score calculated？ Is it at the end of the AL loop?**
>
> We take the best F1 score during each AL loop.
>
> **Q6: How do you calculate individual F1 score for the single-annotation model?  Does the single-annotation model have access to the information about the label’s annotators?**
>
> The single-annotation model does not have access to the annotators id during the training.
> The goal of the experiment is to confirm **how large the gap is** between explicit annotator modelling (annotator-heads) and implicit modelling (training on raw annotation).
>
> Previous work [1] only compared the multi-head model with a single-head model that was trained on the majority vote on the majority vote F1 metrics (where the multi-head model has access to annotator id while the single-head model does not) . Therefore, we want to know if a single-head model trained on the raw annotation can match the performance of the multi-head model on individual F1 metric.
>
> We think this is an important baseline to compare to (Reviewer 2 explicitly asks for this comparison), which tells us how much benefits we can get by explicitly modelling the annotators.
>
> [1] [Dealing with Disagreements: Looking Beyond the Majority Vote in Subjective Annotations](https://aclanthology.org/2022.tacl-1.6) (Mostafazadeh Davani et al., TACL 2022)

---

### Official Review · Reviewer_vzgW · 2023-08-03

**Soundness:** 2

**Excitement:**

3: Ambivalent: It has merits (e.g., it reports state-of-the-art results, the idea is nice), but there are key weaknesses (e.g., it describes incremental work), and it can significantly benefit from another round of revision. However, I won't object to accepting it if my co-reviewers champion it.

**Paper Topic And Main Contributions:**

The paper seeks to improve the model performance when trained on human-annotated data, with multiple annotators per sample. It builds on existing work that shows that exposing the model to annotator uncertainty (instead of aggregated labels) improves downstream performance.

The authors show that combining this setup with an active learning method gets the best of both worlds - a model that is capable of taking uncertainty into account and thus performs better, as well as being sample efficient.

**Reasons To Accept:**

1. The paper does show that the active learning scenario does benefit from the head-per-annotator setup shown to work for regular fine-tuning.

2. The authors do conduct experiments w.r.t different settings(e.g. dense vs sparse multi-annotations, different acquisition functions, different AL methods, etc)

**Reasons To Reject:**

1. The paper ultimately details a report that combines tools already presented in existing work. The acquisition functions, the active learning approaches as well as the idea to use annotator-specific heads to model disagreement are all present in the existing literature. Hence, the paper ultimately presents a new application of existing tools.

2. I also feel that the paper in its current state is missing important baselines (such as just AL without head-per -annotator setup).

**Reproducibility:**

4: Could mostly reproduce the results, but there may be some variation because of sample variance or minor variations in their interpretation of the protocol or method.

**Reviewer Confidence:**

3: Pretty sure, but there's a chance I missed something. Although I have a good feel for this area in general, I did not carefully check the paper's details, e.g., the math, experimental design, or novelty.

---

> ### Author Rebuttal · Authors · 2023-08-28
>
> **Q1: The paper ultimately details a report that combines tools already presented in existing work. The acquisition functions, the active learning approaches as well as the idea to use annotator-specific heads to model disagreement are all present in the existing literature.**
>
> A1: No, we would like to clarify that the acquisition functions we proposed are new. Traditional active learning acquisition functions only consider the output from a single classification head, to which we compare, please see the next answer. Our method explores how to connect active learning and annotator-specific head modelling and design new active learning acquisition functions for that. Therefore, we propose different acquisition functions in two main categories: **group level** and **individual level**. This enables the AL algorithm to specifically query the individual annotator's annotation which previous AL methods cannot achieve.
>
> **Q2: Missing baselines (just AL without head-per-annotator setup)**
>
> A2: We do have included this baseline for comparison. In Figure 2, we compare the multi-head model approach (Multi.) with the single-head  model (without head-per-annotator setup) with 4 different acquisition functions (Ent., DAL, BALD, Rand.). In the upper row, we compare to a  single-head model that learns only from the majority vote. In the bottom row, we compare the multi-head model with the single-head model that leanrs from the raw annotations.
>
> We show that **explicitly modelling each annotator** substantially benefits the learning in terms of majority  vote F1 score, uncertainty estimation and individual annotator modelling.

---

### Official Review · Reviewer_LA7Z · 2023-08-11

**Soundness:** 3

**Excitement:**

3: Ambivalent: It has merits (e.g., it reports state-of-the-art results, the idea is nice), but there are key weaknesses (e.g., it describes incremental work), and it can significantly benefit from another round of revision. However, I won't object to accepting it if my co-reviewers champion it.

**Paper Topic And Main Contributions:**

This paper explores to incorporate annotator disagreement in the active learning process. Based on the experiments on two subjective NLP tasks, it shows that modelling hard labels from each annotator through a multi-head layer could improve the performance with a smaller annotation budget.

**Reasons To Accept:**

It is an interesting and novel direction in active learning. The authors conduct comprehensive evaluation and analysis.

**Reasons To Reject:**

More experiments on other LLMs to justify the conclusions.

**Reproducibility:**

4: Could mostly reproduce the results, but there may be some variation because of sample variance or minor variations in their interpretation of the protocol or method.

**Reviewer Confidence:**

3: Pretty sure, but there's a chance I missed something. Although I have a good feel for this area in general, I did not carefully check the paper's details, e.g., the math, experimental design, or novelty.

---

> ### Author Rebuttal · Authors · 2023-08-28
>
> Thanks for the feedback and your acknowledgement of the novelty of our method.
> We appreciate the suggestion to conduct additional experiments on other language models (LLMs) to further strengthen the conclusions of our study.
> We follow the model choice (BERT) in the prior works in active learning [1] and learning from disagreement [2].
> Given that the BERT/RoBERTa are the most common model choices for text classification and the similar architecture choice between BERT and RoBERTa, we only test our method on BERT.
> We will include RoBERTa model result in the camera-ready version
>
> [1]: [Active Learning for BERT: An Empirical Study](https://aclanthology.org/2020.emnlp-main.638) (Ein-Dor et al., EMNLP 2020)
>
> [2] [Dealing with Disagreements: Looking Beyond the Majority Vote in Subjective Annotations](https://aclanthology.org/2022.tacl-1.6) (Mostafazadeh Davani et al., TACL 2022)

---

### Meta-Review · Area_Chair_jQDG · 2023-09-09

**Recommendation:** 4

**Metareview:**

This paper describes an active learning approach in the case of annotator disagreement. It takes the multihead approach that's now popular in passive learning, and experiment with both known and novel (eg Vote/Mix) data selection procedures. They find that active learning is effective, across three datasets.

On the positive side, this is a timely paper addressing an important problem, and the proposed approach is sensible and works reasonably well.

On the negative side, it is unfortunate that there is not a clear winner between group-level and individual-level annotation.

---

### Decision · Program_Chairs · 2023-10-07

**Decision:**

Accept-Main

**Comment:**

This paper describes an active learning approach in the case of annotator disagreement. It takes the multihead approach that's now popular in passive learning, and experiment with both known and novel (eg Vote/Mix) data selection procedures. They find that active learning is effective, across three datasets.

On the positive side, this is a timely paper addressing an important problem, and the proposed approach is sensible and works reasonably well.

On the negative side, it is unfortunate that there is not a clear winner between group-level and individual-level annotation.